# Quinoa Snack Production at an Industrial Level: Effect of Extrusion and Baking on Digestibility, Bioactive, Rheological, and Physical Properties

**DOI:** 10.3390/foods11213383

**Published:** 2022-10-27

**Authors:** Karen Sofia Muñoz-Pabon, Diego Fernando Roa-Acosta, José Luis Hoyos-Concha, Jesús Eduardo Bravo-Gómez, Vicente Ortiz-Gómez

**Affiliations:** 1Facultad Ciencias Agrarias, Departamento de Agroindustria, Universidad del Cauca, Sede Las Guacas, Popayán 190002, Colombia; 2GIEPRONAL Research Group, School of Basic Sciences, Technology and Engineering, National University Open and Distance (UNAD), Bogotá 110311, Colombia

**Keywords:** gluten-free quinoa snack, functional foods, phenolic compounds, carotenoids, textural properties, digestible starch, quinoa protein digestibility

## Abstract

This research aimed to produce gluten-free snacks on a pilot scale from quinoa flour. These snacks experienced an extrusion process, followed by baking. The effects of these technological processes on carbohydrate and protein digestibility, extractable phenolic compounds (EPP), hydrolyzable phenolic compounds (HPP), antioxidant capacity, and physical properties were evaluated in raw quinoa flour and extruded snacks. Extrusion increased digestible starch (RDS) from 7.33 g/100 g bs to 77.33 g /100 g bs. Resistant starch (RS) showed a variation of 2 g/100 g bs. It is noteworthy that protein digestibility increased up to 94.58 g/100 bs after extrusion and baking. These processes increased HPP content, while EPP and carotenoid content decreased. The samples showed significant differences (*p* < 0.05) in the antioxidant properties determined through the DPPH and ABTS methods. Values of 19.72 ± 0.81 µmol T/g were observed in snacks and 13.16 ± 0.2 µmol T/g in raw flour, but a reduction of up to 16.10 ± 0.68 µmol T/g was observed during baking. The baking process reduced the work of crispness (Wcr) from 0.79 to 0.23 N.mm, while the saturation (C*) was higher in baked ones, showing higher color intensity. The baking process did not influence the viscosity profile. The results in this study respond to the growing interest of the food industry to satisfy consumer demand for new, healthy, and expanded gluten-free snacks with bioactive compounds.

## 1. Introduction

Celiac disease, which affects approximately 0.7% of the world’s population, develops in genetically susceptible individuals who, in response to unclear environmental triggers, develop an immune response triggered by the ingestion of gluten from wheat, barley, rye, and other cereals [1]. Celiac disease can damage the intestinal villi, leading to malnutrition, loss of bone density, infertility and miscarriage, lactose intolerance, and an increased risk of developing various forms of cancer, including intestinal lymphoma and small bowel cancer [2,3]. There is no known cure for celiac disease, and the only treatment is to follow a gluten-free diet. Pseudo-cereals (e.g., amaranth, quinoa, and buckwheat, along with other minor cereals) are a good alternative for being a good source of carbohydrates, protein, dietary fiber, vitamins, and polyunsaturated fatty acids. These grains contain amounts of fiber that are higher than those of other plant foods and cereals and approximately the same as those of wheat [4]. People with celiac disease face a limited supply of gluten-free products with high nutritional value and sensory acceptability [5].

According to [6], extrusion is a suitable process to produce snacks for patients with celiac disease, as starch provides a desirable expanded structure in the product.

An effective approach to increase the supply of food applications based on Andean grains such as Quinoa is to process them into value-added foods. Among the processing technologies to change the functional properties of food ingredients, extrusion is used for cereal-based ingredients. Most snacks are based on blends of gluten-free starches, proteins or flours, dietary fiber materials, and other ingredients that offer a food alternative for people with celiac disease and those interested in healthy foods. These snacks are manufactured in a variety of products with different textures, shapes, and nutritional properties, thanks to extrusion technology that allows mixing, changing, restructuring, and shaping [7]. Quinoa, Chenopodium quinoa Willd., has been cultivated in the Andes for 7000 years. In Colombia, it is grown in the Departments of Nariño, Valle del Cauca, Quindio and Cauca, in the latter for 2018 1507 Ha of the variety Blanca de Jericó are reported [8].It is an amaranthaceous plant very resistant to extreme environmental conditions and has been called a pseudocereal because of its botanical structure and its balanced nutrient composition of proteins from 12.5% to 16.7%, carbohydrates from 60% to 74.7%, and lipids from 5.5% to 8.5% in dry weight, an aspect of interest in the formulation of foods [9]. Previous research [10,11] evaluated the suitability of quinoa as an alternative for the formulation of gluten-free foods.

Considering human nutritional needs, quinoa protein has an excellent amino acid profile, which includes adequate concentrations of essential amino acids necessary for the growth and maintenance of metabolic activities with desirable bioavailability, has high amounts of glutamic acid and aspartic acid, and lower levels of proline and arginine than other cereals [12]. The in vitro digestibility of quinoa crude protein is between 70% and 77% [13]. To increase protein and starch digestibility, quinoa-based foods need thermal processing, and extrusion, cooking, and drying are viable options. For example, according to [14], samples of precooked and drum-dried quinoa showed a high digestibility of starch and protein; extruded samples reached 86%.

Combining heat treatment processes such as cooking, autoclave steam (steam pressure heating), extrusion, drum drying, and microwave heating have been used in cereals to improve palatability, digestibility, stability, inactivation of microorganisms, enzymes, and anti-nutritional compounds and have also been used to increase the shelf-life of foods [15]. However, degradation reactions are accelerated with temperature, so heat treatments may affect nutritional compounds present in quinoa with health-related effects, such as vitamins, pigments, polyphenols, or antioxidant compounds [16].

Quinoa’s abundant phenolic components are a set of bioactive compounds that have health-promoting characteristics [17] and may act as useful antioxidants to reduce the risk of various lifestyle-related diseases such as atherosclerosis and coronary heart disease [18] both related to low-density lipoprotein oxidation. Bioactive components of quinoa have also been reported to protect against certain types of cancer.

Extrusion has shown the extractability of phenolic compounds in quinoa, and [19] found that extrusion influenced the release of bound phenols from quinoa and the best extrusion temperature was 160 °C. According to [20], the effect of heat treatments on phenolic compounds in matrices from cereals is attributed to the type of grain, its nature and location, and the severity and duration of heat treatment is key in changing these compounds.

These processes cause a series of physical and chemical changes due to starch gelatinization, protein denaturation, component interactions, and browning reactions, resulting in improved organoleptic properties, increased nutrient availability, improved antioxidant properties, and the inactivation of heat-labile toxic compounds and enzyme inhibitors [21]. These processes can also alter the total content of phenols, flavonoids, polyunsaturated fatty acids, and the antioxidant activity of foods [22]. Interactions between phenolic compounds, proteins, and starch influence the antioxidant capacity, protein, and starch digestibility, or functional properties of the food [23].

The objective of this study was to develop a gluten-free snack option made with 100% quinoa flour at an industrial level, with a good protein level, and to evaluate the effect of extrusion and baking on its bioactive and digestive properties. In addition, we aimed to present a functional snack option to consumers and propose innovations for future products.

## 2. Materials and Methods

### 2.1. Materials

The milled quinoa was provided by SEGALCO company, Jamundi, Colombia to which toasting was performed, resulting in a final temperature of 98 °C ± 2, to remove saponins. Diammonium salt of 2,2′-azino-bis-(3-ethylbenzothiazoline-6-sulfonic acid) (ABTS) and 2,2-Diphenyl-1-picrylhydrazyl were purchased from Thermo Fisher Scientific (Waltham, MA, USA). The 95% ethanol was obtained from Commercial Outsourcing (Manizales, Colombia). The rest of the reagents and standards were purchased from Sigma Aldrich (Steinheim, Germany), and digestibility kits were supplied by Megazyme (International Ireland Ltd., Wicklow, Ireland).

### 2.2. Snack Production

Before extrusion, the quinoa undergoes an abrasion process on the grain and the seed coats are removed, generating a coproduct rich in saponins and a polished grain free of saponins. The saponin content in quinoa was 10 mg/100 g of polished grain. This value is very low and is not considered anti-nutritional or toxic [24].

The quinoa snacks were made in an industrial twin-screw extruder (CY65-II TWN SCREW EXTRUDER, Qingdao, China). Quinoa flour was humidified up to 18% by spraying the amount of water and mixing continuously in a horizontal mixer with a capacity of 80 kg, then the samples were packed in polyethylene bags and stored for 5 h at room temperature to equilibrate the humidity. Extrusion cooking was carried out with a temperature profile of 75 °C, 105 °C, and 135 °C, the screw rotation speed was 251–253 rpm, and the diameter of the three nozzles was 2.6 mm. Snacks were dried at 170 °C in a tumbler at a speed of 60–65 rpm. The moisture contents of the raw, extruded, and baked snack mixes were 7.7, 7.1, and 2.2%, respectively. Samples were stored in sealed plastic bags at room temperature. Prior to analysis, the extrudates and/or baked snacks were milled with a laboratory grinder (A 11 basic Molino IKA, Germany). Raw quinoa flour (RQF), extruded snacks (ES), and extruded and baked snacks (EBS) were compared. The proximate composition (protein, lipids, dietary fiber, ash, and moisture) of extruded and baked snacks was determined according to the methods proposed by the AOAC (Association of Official Analytical Chemists, 1998) and the carbohydrate content was estimated by the difference [100—(ash + protein + lipids + dietary fiber)]. The EBS presented a proximal composition of 13.7% protein, 3.3% lipids, 9.9% fiber, 3.1% ash, and 78.9% carbohydrates.

### 2.3. In Vitro Digestibility of Starch

The determination of rapidly digestible starch RDS, slowly digestible starch (SDS), fully digestible starch (TDS), and resistant starch (RS) was performed using the Megazyme K-DSTRS assay kit, following the methodology suggested by the supplier (Megazyme International Ireland Ltd., Wicklow, Ireland).

### 2.4. Protein In Vitro Digestibility

Protein digestibility determination was performed using the Megazyme K-PDCAAS assay kit, following the methodology suggested by the supplier (Megazyme International Ireland Ltd., Wicklow, Ireland).

### 2.5. Phenolic Compounds

#### 2.5.1. Extractable Phenolic Compounds (EPP)

EPP extraction and analysis were based on the method described in [25] with slight modifications. First, 2 g ± 0.0500 g of the sample sieved to a size of 100 μm was weighed. The first extraction was performed with ethanol/H_2_O (80/20, more than 1% formic acid), then a second extraction was performed with 8 mL of acetone/H_2_O (70/30).

For the Folin–Ciocalteu reaction, 40 µL of the extract was mixed with 1 mL of the Folin–Ciocalteu reagent (1:4 dilution), 900 µL of H_2_O, and 1 mL of 7.5% (*w*/*v*) Na_2_CO. The sample was incubated for 1 h at room temperature in the dark, and spectrophotometric determination was performed at 765 nm in a UV Vis spectrophotometer (GENESYS™ 10S Thermo Scientific, USA). The total EPP content was expressed as mg gallic acid (GA)/g dry matter equivalent. All analyses were performed in triplicate.

#### 2.5.2. Hydrolyzable Phenolic Compounds (HPP)

HPP extraction and analysis were based on the methods described by Pico et al. [25], with slight modifications. First, 0.8 g ± 0.0050 g of the EPP sediment sample was weighed into a glass tube and 10 mL of methanol/H_2_SO_4_ (90/10) was added. This mixture was left for 22 h at 85 °C with magnetic stirring. The sample was then centrifuged at 3500 rpm for 5 min (Megafuge tabletop centrifuge st4 plus, general purpose, Thermo Scientific, USA) and the supernatant was made up to 25 mL with deionized water. The Folin–Ciocalteu reading was performed as described in step 2.5.1.

### 2.6. Antioxidants

EPP extracts were used to measure antioxidant properties.

#### 2.6.1. ABTS

In the test tube, 4 mL of the ABTS solution was placed and completely covered with aluminum foil. To start the process, 135 μL of standard solution was added and then vortex mixed for 5 s. The reagent blank consisted of 4 mL of acetate buffer and 135 μL of ethanol. The zero point was mixed with 4.5 mL of the ABTS solution and 135 μL of ethanol. The test tube was closed and allowed to react for 30 min to finally measure the absorbance at a wavelength of 729.7 nm.

#### 2.6.2. DPPH

In a test tube, 3.9 mL of the DPPH solution and 100 μL of standard solution were applied to initiate a reaction by vortex stirring for 5 s. Reagent blanks (control) were carried out with ethanol. The zero point was adjusted by adding 3.9 mL of the DPPH solution and 100 μL of ethanol. The sample was covered for 30 min to initiate the reaction, then absorbance was measured at 517 nm.

### 2.7. Determination of Carotenoid Content by Spectrophotometric Method

The extraction of carotenoid pigments was performed following the methodology described by [26,27,28] and modified in two stages: Solid–liquid extraction and liquid–liquid extraction, of which the first extraction stage is adjusted.

Regarding solid–liquid extraction, we dissolved 1 g of the sieved sample up to 100 μm in a falcon tube covered with aluminum foil, with 5 mL of analytical grade acetone, vortexed it for 30 s, and let it rest for 10 min. We added 5 mL of petroleum ether (98%) and 0.1% m/m of butylhydroxytoluene (BHT) and mixed for 1 min. Subsequently, 2.5 mL of distilled water was added and vortexed for 15 s. The sample was centrifuged at 3000 rpm for 10 min, separating the organic phase from the aqueous phase. The organic phase was taken with a graduated pipette and transferred to a falcon tube covered with aluminum foil. The aqueous phase was centrifuged again with 2.5 mL of acetone, 2.5 mL of petroleum ether, 0.1% BHT antioxidant, and 1.25 mL of distilled water. This procedure was performed twice.

Liquid–liquid extraction. The organic phase obtained in the solid-liquid extraction was mixed with 5 mL of sodium chloride (NaCl) with a purity of 100% 0.1 N, vortexed for 15 s, and centrifuged at 3000 rpm for 10 min, obtaining an organic phase and an aqueous phase. The aqueous phase was centrifuged again with the addition of 5 mL of NaCl, and the procedure was carried out in duplicate. Subsequently, a 5 mL volume of the organic phase was transferred to a 100 mL separatory funnel lined with aluminum foil and brought to a volume of 10 mL with petroleum ether and 0.1% BHT antioxidant.

Identification and quantification of β-carotenes: Identification was performed by spectral scanning in a wavelength range from 350 to 600 nm according to the methodology used by Popova et al., 2015. According to [24], the absorption spectrum of β-carotene using petroleum ether as a solvent is 450 nm in a UV Vis spectrophotometer (GENESYS™ 10S Thermo Scientific, USA).

### 2.8. Pasting Properties

The rheological properties of each dispersion were determined using a rheometer (TA INSTRUMENTS, AR 1500, New Castle, DE, USA), equipped with a starch sticking cell, according to the methodology of [29]. Peak viscosity (PV), viscosity drop (BD), and setback (SB) were recorded in duplicate. Finally, using the Savistky–Golay function, the data were smoothed in GraphPad Prism version 5.

### 2.9. Color

The color of the extrudate was determined with a Konica Minolta CM-5 spectrophotometer, controlled by SpectraMagic NX software, with a D65 illuminant and a 10° observation angle. Based on the methodology in [29], chroma (C*), h, and ∆E were calculated with the following equations:(1)$$C*=\sqrt{(a^{*2}+b^{*2}){}^{}}$$
(2)$$h=\tan^{-1}\;\frac{b^{*}}{a^{*}},\;\mathrm{for}\;a^{*}>0;\;b^{*}>0$$
(3)$$h=180+\tan^{-1}\;\frac{b^{*}}{a^{*}},\;\mathrm{for}\;a^{*}<0;\;b^{*}>0$$
(4)$$∆E=\sqrt{{\left(L^{*}-L^{*}{}_{0}\right)}^{2}+{\left(b^{*}-b^{*}{}_{0}\right)}^{2}+\left(a^{*}-a^{*}{}_{0}\right)}$$

### 2.10. Textural Properties

The mechanical properties of the extrudates were analyzed with a texture analyzer (Shimadzu EZ TEST SM, model 500N-168, Japan) following the method of [29,30], with slight modifications. Twenty-five extrudates of each treatment were equilibrated using a moisture chamber (KBF-S 240, Binder, Germany) under 30% RH for 94 h. The area under the curve (S; N.mm) and the number of peaks (n) above 1.5 N were obtained from the force–strain curves and used to calculate the spatial frequency of ruptures (Nsr), the average crushing force (Fcr), and the work of crunching (Wcr), using the following equations:(5)$$\;\mathrm{Nsr}\;\left({\mathrm{mm}}^{-1}\right)=n/d$$
(6)$${\;F}_{\mathrm{cr}}\;\left(N\right)=S/d$$
(7)$$\mathrm{Wcr}\;\left(N.\mathrm{mm}\right)=F_{\mathrm{cr}}/\mathrm{Nsr}$$
donde d = distancia de recorrido de la sonda (mm).

### 2.11. Statistical Analyses

Unless otherwise indicated, experiments were performed in triplicate. Data were expressed as the mean ± standard deviation. Statistical differences were determined by one-way analysis of variance (ANOVA) with a post hoc (*p* < 0.05) multiple range significant difference (Tukey’s) test using Minitab 20.0 Graphs were generated in Graphpad Prism 5.0.

## 3. Results

Table 1 shows the values of RDS, SDS, and RS. The ratio of RDS and RS in RQF, ES, and EBS samples was 7.33 and 2.18, 77.33 and 1.07, and 73.61 and 0.09 The SDS values were 58.51, 2.08, and 2.72 g/100 g for the RQF, ES, and EBS samples g/100 g dry sample. Extrusion and baking affected starch digestibility (*p* < 0.05).

Table 1 shows that the EPPs in the RQF sample of 1.79 mg AG/g decreased to 0.81 mg AG/g after extrusion and baking, with a decrease rate higher than 100%. HPPs and RQF samples of 4.58 mg AG/g increased to 6.98 mg AG/g in EBS, and the rate of increase was higher than 52%. Extrusion and baking had no significant differences for EPP and HPP.

Raw quinoa flour showed lower antioxidant activity and thermal processes led to significant variation (*p* < 0.05). Carotenoid content decreased to 11.33, 8.39, and 7.40 µg β-carotene/g in the raw, extruded, and baked samples (Table 1).

Extrusion increased DPPH scavenging activity by 37% for raw quinoa (3.87 μmol TE/g), while baking had no significant effect on the extrusion process. For activity measured through ABTS, extrusion increased by 49.8% compared to raw, but baking affected antioxidant activity, reducing it by 22.4% regarding ES.

Table 2 shows the viscosity profile of RQF, ES, and EBS. In the samples analyzed, there was no effect of the thermal processes on the cold viscosity (*p* > 0.05), and the hot viscosity values of ES and EBS decreased compared to RQF. As shown in Figure 1, viscosity and temperature values are lower for extruded starch.

The color analysis presented in Table 3 showed values of 86.65, 64.15, and 65.70 for L*, − 0.16, 4.13, and 4.59 for a*, and 15.06, 18.81, and 20.24 for b* in the raw, extruded, and baked samples. The highest value of L* was observed in the raw quinoa sample, showing that the extrusion process decreased the lightness. As for redness, the highest a* value was obtained in the baked snacks. The b* value, which measures the yellowish degree, was highest in the baked snacks. The total change (∆E), which allows for a color comparison with raw quinoa flour, ranged from 22.06 for extruded and baked snacks to 23.23 for extruded-only snacks.

As shown in Figure 2, baking has a significant effect *p* < 0.05 on the texture of the snack, as the ES samples presented a hardness of 7.1 N m, while for EBS it was 5.74 N. The Nsr increased in the baked samples above 100% compared to the ES, showing a higher degree of crunchiness in the EBS. Fcr and Wcr presented higher values in the ES extruded samples, being 3.62 and 0.79, respectively, while for the EBS, values of 2.79 and 0.23 for Fcr and Wcr, respectively, were presented.

## 4. Discussion

### 4.1. Carbohydrate and Protein Digestibility

The amount of RDS in raw samples is lower than in extruded and/or baked snacks because more RDS was released in baked samples, which has dispersed starch and is found in starchy foods that have undergone a cooking process, which is also the starch fraction that causes a sudden increase in blood glucose levels after ingestion [31]. Unlike raw flour, ES and EBS increased the release of RDS because extrusion generates stabilization in the snacks and thus makes them less prone to retrogradation, as corroborated by the low amount of resistant starch. Extrusion can change the physicochemical characteristics of cereals, generating gelatinization, due to the combination of humidity and high temperatures, which causes hydrolysis in the starch chains, increasing the glycemic index [32,33].

Resistant starch has been defined by EURESTA (European Food Linked Agro-Industrial Research Concerted Action on Resistant Starch) as the “total amount of starch and starch degradation products that resist digestion in the small intestine of healthy individuals”. This resistance is related to the association between starch polymers, therefore a higher amylose content in starch is associated with slower digestibility values [31].

According to [34], quinoa, amaranth, and millet have an amylose content of 10.92%, 17.96%, and 12.77%, respectively, i.e., the starch present in quinoa with lower amylose content is naturally less resistant to digestion. The extruded samples present a lower RS content, due to the action of shearing and heat that alters the molecular structure of quinoa starch increasing the amount of amylose and decreasing the crystalline region [35], making it more susceptible to hydrolysis mediated by the action of digestive enzymes, behavior that coincides with the results of other researchers [36].

In raw samples, the SDS values are significantly higher because the raw flour has not been subjected to a transformation or hydrolysis process, so in these samples, the SDS is found as inaccessible amorphous starch with a crystalline structure of type A and type C in cereals. In snacks, the SDS fraction is type B in the form of granules or retrograded, and this SDS fraction is completely digested in the small intestine, more slowly than the RDS fraction [31].

The high extrusion temperatures of 105 °C in this study changed the secondary structure of the quinoa protein, causing increased susceptibility to digestive enzymes and improved solubility [12]. Protein digestibility values of raw quinoa flour were reported to be above 70% [13,37], higher than other grains, because all essential amino acids in quinoa protein have sufficient levels of sulfur amino acids, cysteine, and are rich in lysine, histidine, and methionine, which are the limiting amino acids in common cereals [12]. The lower digestibility in raw quinoa compared to extruded quinoa may also be related to the fact that quinoa seed has a starch formed by amylose and amylopectin that can influence protein digestion, absorbing gastric liquid and hindering the entry of acid and pepsin into the protein [38].

In comparison with extrudates, similar protein digestibility values were obtained by [39]. According to [40], in comparison with other cereals, the best protein quality was found in quinoa seeds, being similar to that of casein, and their results were based on the amino acid profile and high apparent digestibility.

### 4.2. Effect of Processing on EPP and HPP

This shows that a dry process with high pressure such as extrusion was effective in the release of hydrolyzable phenolic compounds bound in quinoa [41] due to the release of some unavailable phenols bound to cell walls, the presence of Maillard reaction products (MRP) including antioxidants, and the improvement of extractability [42]. Once released, strong antioxidant, anticancer, anti-inflammatory, antiobesity, antidiabetic, and anti-central nervous system disease effects can be expected by inbound phenols [43,44]. Previous authors [45] suggest that the combination of low temperatures (<140 °C) and low humidity (<14%) can keep higher phenolic contents and enhance antioxidant activity.

Similar results of the effect on extrusion-based phenolics were also reported for extruded quinoa [19].

### 4.3. Effect of Processing on the Antioxidant Activities and Carotenoids

The increase in phenolic compounds and elevation of antioxidant activities in quinoa is related to the formation of Maillard reaction products (MRPs), and high-pressure baking of quinoa improved the antioxidant potential by the release of conjugated phenolic compounds by breaking the complex structures [16]. The loss of antioxidant activity caused by cooking is unfavorable due to the possibility of inducing oxidative decomposition by thermal degradation of phenolic compounds [15].

Thermal decomposition is the most likely cause for these compounds because of their chemical structure with unsaturated covalent bonds that are more prone to degradation [46]. According to the literature, the degradation of carotenoid pigments is higher in an oily medium containing high proportions of unsaturated fatty acids [46]. Quinoa grain has 19–12.3% total saturated lipids such as palmitic acid; 25–28.7% total monounsaturated lipids such as oleic acid; and 58.3% total polyunsaturated lipids such as linoleic acid (approximately 90%) [47]. Under extrusion temperatures, these can be oxidized [48]. It is also possible that lipids in the food matrix react with carotenoids, because of their protective effect against oxidation [49].

According to [50], quinoa seeds are a rich source of flavonoids, consisting mainly of quercetin and kaempferol glycosides, as well as tannins, ferulic acid, p-coumaric acid, and caffeic acid, suggesting that quinoa could represent an important source of free radical inhibition. Previous research [50] reports that, in quinoa, the extrusion process increases the antioxidant activity due to the increase in phenolic compounds released during thermal processing, where changes occur such as the polymerization and oxidation of phenolics, thermal degradation, the depolymerization of high-molecular-weight compounds, such as condensed tannins, and the production of MRPs. Furthermore, different environmental conditions, cultivation techniques, or genetic factors can influence the phenolic contents.

Extrusion processing resulted in a decrease in free phenolic acids, EPP, and carotenoids and an increase in bound phenolic acids, HPP, and the antioxidant effect. It is important to mention that sugars present in the snacks may react with the Folin–Ciocalteu reagent and contribute to the total absorbance, which may overestimate the results of phenolic compounds [15].

### 4.4. Pasting

The viscosity and temperature values are lower for extruded starch because molecular degradation shows a reduction in starch size [51]. This could result from starch damage and the lower tendency of molecules to tightly pack during cooling.

Therefore, smaller polymers have a higher number of hydroxyl groups and a higher affinity to form hydrogen bonds with water molecules; during cooling, reassociation between amylose molecules results in making a gel structure and leads to an increase in the cold viscosity of the extrudates [51]. The pasting temperature shows the minimum temperature at which starch gelatinizes, known as the gelatinization temperature. In extrudates, the pasting temperature is lower than in crude, because the gelatinization and structural degradation of starch during the extrusion process generates a lower swelling capacity so that the starch granule gelatinizes earlier. Previous researchers [52,53] obtained a similar trend in millet-based extruded products.

“Setback” values are indicators of the retrogradation and rearrangement of starch molecules during the cooling process [51]. In ES and EBS samples, setback values decreased after extrusion, showing that the extruded samples were less prone to retrogradation, and were a more resistant starch formation as corroborated during starch digestibility simulation. In the raw samples, more RS was formed from the starchy material in the presence of moisture, which may act as a plasticizer facilitating retrogradation, an aspect that may contribute to decreased glycemic [45].

If the final viscosity and setback of quinoa starch are compared with potato or wheat starch, it can be reported that after a heating process in quinoa flour, retrogradation and final viscosity decrease [54] making it a more stable starch, because the starch present in quinoa has a lower amylose content (7–27%) and its size can vary between (0.5–3 μm in diameter) [55], characteristics that make it less prone to retrogradation. This is an aspect of snack production that can contribute to the preservation of greater stability of the food matrix during storage.

### 4.5. Color

Accordingly, [51] reported that in mixtures of raw and extruded cereals, most of the population perceives a total color difference greater than 3, and if the color difference is greater than 6, the colors are in different color groups. Chroma (C*) is considered a quantitative color characteristic. Therefore, baked snacks present a higher color intensity, with a value greater than 20.75. The hue angle (h*) is a qualitative characteristic of color. A lower hue angle shows a more yellow character, with EBS with a value of 77.22 presenting the lowest value.

Color changes are affected by thermal processes that may produce Maillard reactions, caramelization, hydrolysis, and pigment degradation [56].

### 4.6. Texture

Texture parameters are key characteristics faced in extruded expanded snacks. Hardness refers to the maximum strength reached by the sample during compression. In these products, attributes with lower density, lower hardness, and a crunchy porous structure are preferred [29]. There are significant differences in the hardness values of ES and EBS, with the latter obtaining a lower hardness value, which may be related to the higher water content of the unbaked snacks. The work of crisping (Wcr) data followed a similar pattern to Fcr. Baking had a significant effect on Wc, implying a lower energy need per crushing peak. Baking also had a significant effect on the spatial frequency of breakage (Nsr). Baked snacks recorded a higher number of sequential fractures and, as a consequence, more crushing peaks in the same probe travel distance.

There is a relationship between moisture and textural properties of foods. A hygroscopic food contains water in bound and unbound forms that can influence its structure and texture [57]. Given that the EBS samples presented lower moisture (2.2%), it can be said that the baking process removes some of the water present in the snack, resulting in crispier foods. Therefore, the intensity of the crispiness depends on the water activity in the dry snack food products [58].

## 5. Conclusions

The extrusion process caused a loss of carotenoids and EPP; however, it allowed the release of HPP and increased antioxidant activity. After the baking process, HPP release was not significantly affected; however, antioxidant capacity and carotenoid content decreased. This suggests that although the baking process improves physical properties such as texture and color, it generates losses of active compounds. Extrusion and baking increased the digestibility of protein and rapidly digestible starch SDR, which is associated with increased blood glucose. Quinoa snacks show an increase in SDR and a decrease in RS, i.e., an opposite trend, which is common in processes involving changes in the molecular structure of starch. These changes are reflected in a decrease in the final viscosity of the extruded samples and a lower rate of retrogradation, a convenient aspect in the stability of the snack during storage. The results of this research suggest new studies and improvements in the industrial processes, preserving the physical, nutritional, and bioactive characteristics, and also suggest measurements of phenolic compounds and amino acids in the food matrix by techniques such as an HPLC chromatogram before and after thermal processes. Quinoa flour is an alternative in the preparation of snacks with good protein quality, antioxidant capacity, better palatability, and high protein digestibility, becoming a gluten-free food option.

## Figures and Tables

**Figure 1 foods-11-03383-f001:**
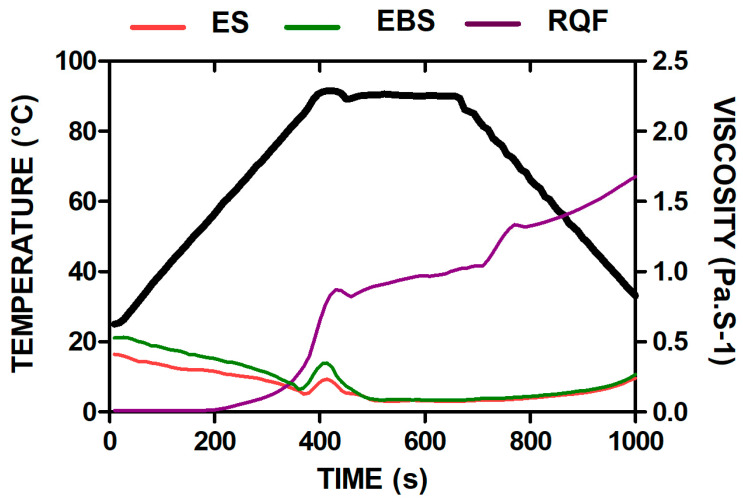
Pasting properties of ES (extruded snack), EBS (extruded and baked snack), and RQF (raw quinoa flour).

**Figure 2 foods-11-03383-f002:**
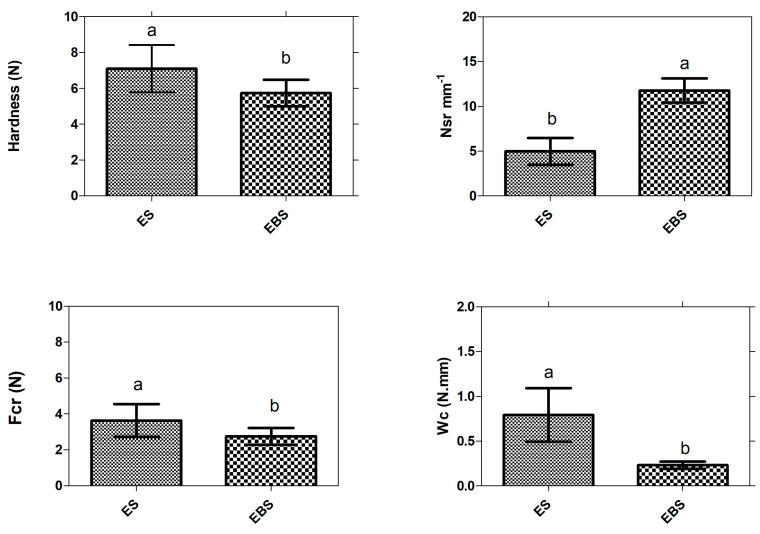
Textural properties of ES (extruded snack) and EBS (extruded and baked snack): Hardness; spatial frequency of ruptures (Nsr); average crushing force (Fcr); and crispness work (Wc). Texture parameter values followed by different lowercase letters, respectively, are significantly different (*p* < 0.05).

**Table 1 foods-11-03383-t001:** Different bioactive and digestibility properties of raw, extruded, and baking quinoa flour products.

Sample	Extractable Phenolic Compounds (EPP) (mg AG/g)	Hydrolyzable Phenolic Compounds (HPP) (mg AG/g)	ABTS (µmol de T/g)	DPPH (µmol de T/g)	Carotenoides µg β-Caroteno/g	RDS g/100 g	SDS g/100 g	TDS g/100 g	RS g/100 g	In Vitro Protein Digestibility g/100 g
RQF	1.79 ± 0.06 ^a^	4.58 ± 0.10 ^b^	13.16 ± 0.20 ^c^	3.87 ± 0.21 ^b^	11.33 ± 0.82 ^a^	7.33 ± 0.00 ^c^	58.51 ± 0.00 ^a^	73.54 ± 0.07 ^b^	2.18 ± 0.03 ^a^	88.51 ± 0.12 ^c^
ES	0.75 ± 0.03 ^b^	7.23 ± 0.78 ^a^	19.72 ± 0.81 ^a^	5.32 ± 0.27 ^a^	8.39 ± 0.69 ^b^	77.33 ± 0.03 ^a^	2.08 ± 0.01 ^b^	79.21 ± 0.00 ^a^	1.07 ± 0.00 ^b^	93.82 ± 0.81 ^b^
EBS	0.81 ± 0.04 ^b^	6.98 ± 1.22 ^a^	16.10 ± 0.68 ^b^	5.30 ± 0.28 ^a^	7.40 ± 0.56 ^c^	73.61 ± 0.01 ^b^	2.72 ± 0.02 ^b^	79.44 ± 0.00 ^a^	0.09 ± 0.01 ^c^	94.58 ± 1.05 ^a^

Values are presented as mean ± SD. For each parameter, different letters indicate significant differences at *p* < 0.05.

**Table 2 foods-11-03383-t002:** Pasting properties of non-extruded and extruded samples.

Sample	Peak Viscosity (Pa.s)	Peak Time (s)	Trouhg (Pa.s)	Final Viscosity (Pa.s)	Pasting Temperature (°C)	Breakdown	Setback (Pa.s)
	Extruded samples						
ES	0.130 ± 0.006 ^a^	1033 ± 7.07 ^a^	0.076 ± 0.003 ^b^	0.19 ± 0.04 ^b^	90.72 ± 0.95 ^b^	0.118 ± 0.04 ^a^	0.183 ± 0.005 ^b^
EBS	0.158 ± 0.004 ^a^	1035 ± 0.72 ^a^	0.085 ± 0.002 ^b^	0.21 ± 0.00 ^b^	91.00 ± 0.00 ^b^	0.127 ± 0.02 ^a^	0.203 ± 0.013 ^b^
	Non-extruded samples						
RQF	0.127 ± 0.02 ^a^	565 ± 35.34 ^b^	1.220 ± 0.32 ^a^	1.13 ± 0.02 ^a^	94.15 ± 1.90 ^a^	0.082 ± 0.08 ^a^	1.125 ± 0.003 ^a^

Values are presented as mean ± SD (*n* = 2). ES (extruded snack) and EBS (extruded and baked snack). For each parameter, different letters indicate significant differences at *p* < 0.05.

**Table 3 foods-11-03383-t003:** Color analysis of raw, extruded, and baked and extruded quinoa flour snacks.

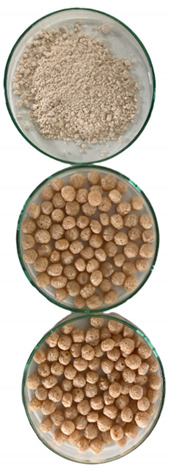	**L***	**b***	**b***	**C***	**h**	**ΔE**
	RQF					
	86.65 ^a^ ± 0.13	−0.16 ^c^ ± 0.01	15.06 ^c^ ± 0.08	15.06 ± 0.08 ^c^	90.61 ± 0.05 ^a^	-
	ES					
	64.15 ^c^ ± 0.17	4.13 ^b^ ± 0.07	18.81 ^b^ ± 0.25	19.25 ± 0.25 ^b^	77.62 ± 0.23 ^b^	23.23
	EBS					
	65.70 ^b^ ± 0.43	4.59 ^a^ ± 0.14	20.24 ^a^ ± 0.49	20.75 ± 0.40 ^a^	77.22 ± 0.31 ^c^	22.06

Values are presented as mean ± SD (*n* = 5). For each parameter, different letters indicate significant differences at *p* < 0.05.

## Data Availability

Not applicable.

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
