# Peer review of "Quinoa Snack Production at an Industrial Level: Effect of Extrusion and Baking on Digestibility, Bioactive, Rheological, and Physical Properties"

_foods, 2022, doi:10.3390/foods11213383_

Round 1
Reviewer 1 Report
manuscript foods-1882280
Title: Quinoa snack production at industrial level: Effect
of extrusion and baking on digestibility, bioactive, rheological and
physical properties
The topic of the article is current and relevant to the industry. Changes occurring during extrusion and roasting of quinoa samples were investigated. The samples developed are gluten-free and suitable for people suffering from celiac disease as well as for people on a high protein diet. The abstract and the methods used for analysis are clear and specific, in the methods, it is good to describe the specifics of the devices with which the analyzes were performed (model, manufacturer). To expand on the comment on starch digestibility and the content and significance of the other starches analyzed. The discussion discusses the amylose content of quinoa but analyzes showing the content in both the raw material and the extruded and baked products are lacking.
To prove the biological value, it is necessary to analyze the protein content and amino acid composition before and after heat treatment. The results for pasting properties in Table 2 to analyze in more depth what information the obtained data give to each indicator.
The conclusion is very extensive, to synthesize and shorten it for clarity.
Author Response
Response to Reviewer 1 Comments
The topic of the article is current and relevant to the industry. Changes occurring during extrusion and roasting of quinoa samples were investigated. The samples developed are gluten-free and suitable for people suffering from celiac disease as well as for people on a high protein diet.
Point 1: The abstract and the methods used for analysis are clear and specific, in the methods, it is good to describe the specifics of the devices with which the analyzes were performed (model, manufacturer).
Response 1: The equipment mentioned present model, manufacturer
Point 2: To expand on the comment on starch digestibility and the content and significance of the other starches analyzed. The discussion discusses the amylose content of quinoa but analyzes showing the content in both the raw material and the extruded and baked products are lacking.
Response 2: The amylose content was not verified in this study, so data from other authors were used.
Point 3: To prove the biological value, it is necessary to analyze the protein content and amino acid composition before and after heat treatment.
Response 3: we agree with the reviewer, however, during the present study no identification of amylose and amino acid contents was performed, for this reason we include this recommendation in the conclusions.
“The results of this research suggest new studies and improvements in the industrial processes, preserving the physical, nutritional and bioactive characteristics, and also suggest measurements of phenolic compounds and amino acids in the food matrix by techniques such as HPLC chromatogram before and after the thermal processes”.
Point 4: The results for pasting properties in Table 2 to analyze in more depth what information the obtained data give to each indicator.
Response 4: we agree with the reviewer, this paragraph is added to improve the explanation
“If the final viscosity and setback of quinoa starch is compared with potato or wheat starch, it can be reported that after a heating process in quinoa flour, retrogradation and final viscosity decrease [49], being a more stable starch, because the starch present in quinoa has a lower amylose content (7-27%) and its size can vary between (0.5-3 μm in diameter) [50], characteristics that make it less prone to retrogradation. An aspect that in snack production can contribute to the preservation of greater stability of the food matrix during storage”.
Point 4: The conclusion is very extensive, to synthesize and shorten it for clarity.
Response 5: the conclusion is changed
“The extrusion process caused loss of carotenoids and EPP, however, it allowed the release of HPP and increased antioxidant activity. After the baking process, HPP release was not significantly affected, however, antioxidant capacity and carotenoid content decreased. This suggests that although the baking process improves physical properties such as texture and color, it generates losses of active compounds. Extrusion and baking increased the digestibility of protein and rapidly digestible starch SDR, which is associated with increased blood glucose. Quinoa snacks show a increase in SDR and a decrease in RS, i.e. an opposite trend, which is common in processes involving changes in the molecular structure of starch. These changes are reflected in a decrease in the final viscosity of the extruded samples and a lower rate of retrogradation, a convenient aspect in the stability of the snack during storage. The results of this research suggest new studies and improvements in the industrial processes, preserving the physical, nutritional and bioactive characteristics, and also suggest measurements of phenolic compounds and amino acids in the food matrix by techniques such as HPLC chromatogram before and after the thermal processes. Quinoa flour is an alternative for the preparation of snacks with good protein quality, antioxidant capacity, better palatability and high protein digestibility, becoming a gluten-free food option.”

Reviewer 2 Report
Abstract:
Line 17 protein digestibility unit must be corrected.
Introduction:
Novelty must be explained.
Materials and methods:
Water absorption index and water solubility index of the products should be given since they are related to starch digestebility.
I should be given the effects of toasting process on products. Is it measured the saponin values after toasting?
How it is decided the production parameters (Moisture, temperature profile of the extruder, screw rotational speed, drying conditions etc.)?
What is the mixture mentioned in Line 104?
Results:
All tables and Figures should be checked for English.
Discussion:
Discussions should be improved and comparisons should be given between the results and with the literature values.
Lines 424-428 should be checked and it should be supported bty the literature.
Conclusions:
Conclusion is not about summarising the key results of this research, and it should highlight the insights.
Author Response
Response to Reviewer 2 Comments
Abstract:
Point 1: Line 17 protein digestibility unit must be corrected.
Response 1: the line was corrected
Introduction:
Point 2: Novelty must be explained.
Response 2: we explain the novelty in this paragraph
“The objective of this study was to develop a gluten-free snack option made with 100% quinoa flour at an industrial level, with a good protein level, and to evaluate the effect of extrusion and baking on its bioactive and digestive properties. In addition to present a functional snack option to consumers and to propose innovations for future products”.
Materials and methods:
Point 3: Water absorption index and water solubility index of the products should be given since they are related to starch digestebility.
Response 3: The values of water absorption index and water solubility index were not performed, however, the suggestion will be taken into account in future studies.
Point 4: I should be given the effects of toasting process on products. Is it measured the saponin values after toasting?
Response 4: the recommendation is accepted, however in this study we seek to measure the effect of extrusion and baking on a ready-to-eat product, the saponin value is measured at the polished kernel. This process is described in the snack production section.
“Before extrusion, the quinoa undergoes an abrasion process on the grain, the seed coats are removed generating a coproduct rich in saponins and a polished grain free of saponins. The saponin content in quinoa was 10mg/100g of polished grain. This value is very low and is not considered anti-nutritional or toxic [21]”.
Point 5: How it is decided the production parameters (Moisture, temperature profile of the extruder, screw rotational speed, drying conditions etc.)?
Response 5: For the design of the 100% quinoa snack product, SEGALCO has a work team dedicated to the design and development of new products, to make the design the company has marketing studies and trends in snacks, also for 10 years the innovation and development team has conducted tests in extrusion process, which has allowed the acquisition of preliminary process variables and mixtures.
Point 6: What is the mixture mentioned in Line 104?
Response 6: The line change to “Quinoa flour was humidified”
Results:
Point 7: All tables and Figures should be checked for English.
Response 7: Figure 2 is corrected to English
Discussion:
Discussions should be improved and comparisons should be given between the results and with the literature values.
Point 9: Lines 424-428 should be checked and it should be supported bty the literature.
Response 9: the Lines 424-428 is corrected according to the reviewer's request.
“Extrusion processing resulted in a decrease of free phenolic acids, EPP, and carotenoids and an increase of bound phenolic acids, HPP, and antioxidant effect. It is important to mention that sugars present in the snacks may react with Folin-Ciocalteu and contribute to the total absorbance, which may overestimate the results of phenolic compounds [12]”.
Conclusions:
Point 10: Conclusion is not about summarising the key results of this research, and it should highlight the insights.
Response 10: the conclusion is changed
“The extrusion process caused loss of carotenoids and EPP, however, it allowed the release of HPP and increased antioxidant activity. After the baking process, HPP release was not significantly affected, however, antioxidant capacity and carotenoid content decreased. This suggests that although the baking process improves physical properties such as texture and color, it generates losses of active compounds. Extrusion and baking increased the digestibility of protein and rapidly digestible starch SDR, which is associated with increased blood glucose. Quinoa snacks show a increase in SDR and a decrease in RS, i.e. an opposite trend, which is common in processes involving changes in the molecular structure of starch. These changes are reflected in a decrease in the final viscosity of the extruded samples and a lower rate of retrogradation, a convenient aspect in the stability of the snack during storage. The results of this research suggest new studies and improvements in the industrial processes, preserving the physical, nutritional and bioactive characteristics, and also suggest measurements of phenolic compounds and amino acids in the food matrix by techniques such as HPLC chromatogram before and after the thermal processes. Quinoa flour is an alternative for the preparation of snacks with good protein quality, antioxidant capacity, better palatability and high protein digestibility, becoming a gluten-free food option”.

Reviewer 3 Report
In the current manuscript, the effects of the technological processes on the carbohydrate and protein digestibility and physicochemical characteristics of quinoa snacks were studied.
The paper is very well designed and written well. It has enough good experiments, too. The results are well presented and discussed. In my opinion, the present manuscript needs minor revisions as follows:
-In the introduction, the global production of quinoa in recent years, which has been growing, should be mentioned by providing data.
L56: write more about the quinoa plant such as scientific name, family, origin, …
Some examples of research on the consumption of quinoa in people with celiac disease should be mentioned. Such as:
Zevallos, V. F., Ellis, H. J., Šuligoj, T., Herencia, L. I., & Ciclitira, P. J. (2012). Variable activation of immune response by quinoa (Chenopodium quinoa Willd.) prolamins in celiac disease. The American journal of clinical nutrition, 96(2), 337-344.
Peñas, E., Uberti, F., di Lorenzo, C., Ballabio, C., Brandolini, A., & Restani, P. (2014). Biochemical and immunochemical evidences supporting the inclusion of quinoa (Chenopodium quinoa Willd.) as a gluten-free ingredient. Plant foods for human nutrition, 69(4), 297-303.
L92: put "." after "properties".
L104: Write the device specifications such as model, manufacturer, etc.
-In the materials and methods section, three different temperatures (75 â—¦C, 105 â—¦C and 135 â—¦C ) for baking are mentioned. Have the effects of baking temperature been investigated separately on product properties? Because higher temperatures will lead to denaturation of proteins, more hydrolysis of fats, and even browning.
-4.3. section: The results of Table 1 show that due to extruder and baking, the amount of antioxidant activity has increased in general, but the antioxidant activity of ABTS in baking has decreased significantly compared to the extruder. Explain more about this.
- Do you suggest mixing quinoa flour with other gluten-free flours or adding some hydrocolloids to improve the properties of the product?
Author Response
Response to Reviewer 3 Comments
In the current manuscript, the effects of the technological processes on the carbohydrate and protein digestibility and physicochemical characteristics of quinoa snacks were studied.
The paper is very well designed and written well. It has enough good experiments, too. The results are well presented and discussed. In my opinion, the present manuscript needs minor revisions as follows:
Point 1: -In the introduction, the global production of quinoa in recent years, which has been growing, should be mentioned by providing data.
Response 1: Quinoa, Chenopodium quinoa Willd., has been cultivated in the Andes for 7000 years. In Colombia it is grown in the Departments of Nariño, Valle del Cauca, Quindio and Cauca, in the latter for 2018 1507 Ha of the variety Blanca de Jericó are reported [8].It is an amaranthaceous plant very resistant to extreme environmental conditions and has been called a pseudocereal because of its botanical structure and its balanced nutrient composition of proteins from 12.5% to 16.7%, carbohydrates from 60% to 74.7%, and lipids from 5.5% to 8.5% in dry weight, an aspect of interest in the formulation of foods [8]. [9], [10] valuated the suitability of quinoa as an alternative for the formulation of gluten-free foods.
Point 2: L56: write more about the quinoa plant such as scientific name, family, origin, …
Response 2: Quinoa, Chenopodium quinoa Willd., has been cultivated in the Andes for 7000 years. It is an amaranthaceous plant very resistant to extreme environmental conditions and has been called a pseudocereal because of its botanical structure and its balanced nutrient composition of proteins from 12.5% to 16.7%, carbohydrates from 60% to 74.7%, and lipids from 5.5% to 8.5% in dry weight, an aspect of interest in the formulation of snacks [8].
Point 3: Some examples of research on the consumption of quinoa in people with celiac disease should be mentioned. Such as:
Zevallos, V. F., Ellis, H. J., Šuligoj, T., Herencia, L. I., & Ciclitira, P. J. (2012). Variable activation of immune response by quinoa (Chenopodium quinoa Willd.) prolamins in celiac disease. The American journal of clinical nutrition, 96(2), 337-344.
Peñas, E., Uberti, F., di Lorenzo, C., Ballabio, C., Brandolini, A., & Restani, P. (2014). Biochemical and immunochemical evidences supporting the inclusion of quinoa (Chenopodium quinoa Willd.) as a gluten-free ingredient. Plant foods for human nutrition, 69(4), 297-303.
Response 3: the references were added
“[9], [10] evaluaron la idoneidad de la quinoa como alternativa para la formulación de alimentos libre de gluten”.
Point 4: L92: put "." after "properties".
Response 4: The objective of this study was to develop a gluten-free snack option made with 100% quinoa flour at an industrial level, with a good protein level, and to evaluate the effect of extrusion and baking on its bioactive and digestive properties. In addition to present a functional snack option to consumers and to propose innovations for future products.
Point 5: L104: Write the device specifications such as model, manufacturer, etc.
Response 5: The quinoa snacks were made in an industrial twin-screw extruder (CY65-II TWN SCREW EXTRUDER, China).
Point 6: In the materials and methods section, three different temperatures (75 â—¦C, 105 â—¦C and 135 â—¦C ) for baking are mentioned. Have the effects of baking temperature been investigated separately on product properties? Because higher temperatures will lead to denaturation of proteins, more hydrolysis of fats, and even browning.
Response 6: different extrusion temperatures were not studied, the three values mentioned refer to the profile used in the extruder which has three heating zones. This ideal temperature profile has been studied and standardized by the SEGALCO company.
Point 7: -4.3. section: The results of Table 1 show that due to extruder and baking, the amount of antioxidant activity has increased in general, but the antioxidant activity of ABTS in baking has decreased significantly compared to the extruder. Explain more about this.
Response 7: The increase in phenolic compounds and elevation of antioxidant activities in quinoa is obeyed to the formation of Maillard reaction products (MRPs), high-pressure baking of quinoa improved the antioxidant potential by the release of conjugated phenolic compounds by breaking the complex structures [15]. The loss of antioxidant activity caused by cooking is unfavorable due to the possibility of inducing oxidative decomposition by thermal degradation of phenolic compounds [14].
Point 8: Do you suggest mixing quinoa flour with other gluten-free flours or adding some hydrocolloids to improve the properties of the product?
Response 8 yes, it is a good suggestion for future studies
Point 9: The conclusion is very extensive, to synthesize and shorten it for clarity.
Response 9: the conclusion is changed
“The extrusion process caused loss of carotenoids and EPP, however, it allowed the release of HPP and increased antioxidant activity. After the baking process, HPP release was not significantly affected, however, antioxidant capacity and carotenoid content decreased. This suggests that although the baking process improves physical properties such as texture and color, it generates losses of active compounds. Extrusion and baking increased the digestibility of protein and rapidly digestible starch SDR, which is associated with increased blood glucose. Quinoa snacks show a increase in SDR and a decrease in RS, i.e. an opposite trend, which is common in processes involving changes in the molecular structure of starch. These changes are reflected in a decrease in the final viscosity of the extruded samples and a lower rate of retrogradation, a convenient aspect in the stability of the snack during storage. The results of this research suggest new studies and improvements in the industrial processes, preserving the physical, nutritional and bioactive characteristics, and also suggest measurements of phenolic compounds and amino acids in the food matrix by techniques such as HPLC chromatogram before and after the thermal processes. Quinoa flour is an alternative for the preparation of snacks with good protein quality, antioxidant capacity, better palatability and high protein digestibility, becoming a gluten-free food option.”

Round 2
Reviewer 2 Report
Current version of the manuscript can be accepted for publication.